# Human Body Performance with COVID-19 Affectation According to Virus Specification Based on Biosensor Techniques

**DOI:** 10.3390/s21248362

**Published:** 2021-12-15

**Authors:** Mohammed Jawad Ahmed Alathari, Yousif Al Mashhadany, Mohd Hadri Hafiz Mokhtar, Norhafizah Burham, Mohd Saiful Dzulkefly Bin Zan, Ahmad Ashrif A Bakar, Norhana Arsad

**Affiliations:** 1Department of Electrical, Electronic and Systems Engineering, Faculty of Engineering and Built Environment, Universiti Kebangsaan Malaysia (UKM), Bangi 43600, Malaysia; p108735@siswa.ukm.edu.my (M.J.A.A.); hadri@ukm.edu.my (M.H.H.M.); hafizahburham@uitm.edu.my (N.B.); saifuldzul@ukm.edu.my (M.S.D.B.Z.); ashrif@ukm.edu.my (A.A.A.B.); 2Department of Electrical Engineering, College of Engineering, University of Anbar, Anbar 00964, Iraq; yousif.mohammed@uoanbar.edu.iq; 3School of Electrical Engineering, College of Engineering, Universiti Teknologi MARA, Shah Alam 40450, Malaysia

**Keywords:** SARS-CoV-2 detection, COVID-19 detection, coronavirus detection, transmission human exchange, next generation sequencing (NGS), RT-PCR, LAMP, biosensor application

## Abstract

Life was once normal before the first announcement of COVID-19’s first case in Wuhan, China, and what was slowly spreading became an overnight worldwide pandemic. Ever since the virus spread at the end of 2019, it has been morphing and rapidly adapting to human nature changes which cause difficult conundrums in the efforts of fighting it. Thus, researchers were steered to investigate the virus in order to contain the outbreak considering its novelty and there being no known cure. In contribution to that, this paper extensively reviewed, compared, and analyzed two main points; SARS-CoV-2 virus transmission in humans and detection methods of COVID-19 in the human body. SARS-CoV-2 human exchange transmission methods reviewed four modes of transmission which are Respiratory Transmission, Fecal–Oral Transmission, Ocular transmission, and Vertical Transmission. The latter point particularly sheds light on the latest discoveries and advancements in the aim of COVID-19 diagnosis and detection of SARS-CoV-2 virus associated with this disease in the human body. The methods in this review paper were classified into two categories which are RNA-based detection including RT-PCR, LAMP, CRISPR, and NGS and secondly, biosensors detection including, electrochemical biosensors, electronic biosensors, piezoelectric biosensors, and optical biosensors.

## 1. Introduction and Overview of Coronaviruses:

The Coronaviridae family to which the Coronavirus belongs is also a part of the Nidovirales order and is the subfamily of Orthocoronavirinae. The subfamily of Orthocoronavirinae consists of four types which are delta (δ) beta (β), alpha (α), and gamma (γ) coronavirus as shown in Figure 1, these viruses share qualities where they are all enveloped, contain single-stranded RNA, have positive-sense and are not segmented viruses that cause minor or critical illnesses in some breathing creatures which includes human beings. The name of Coronavirus is driven from its club-shaped spikes, protruding from the virion surface resulting in a solar corona shape [1,2,3].

Preliminary studies on Coronaviruses were reported in the 1930s. In contrast, the human type of the virus named (HCoVs) was first discovered in the 1960s. Until recently, seven types of the virus have been discovered including the HCoVs-NL63 and HCoVs-229E 229E (α-Corona viruses) and HCoVs-OC43, HCoVs-HKU1 (β-Corona viruses), severe acute respiratory syndrome-CoV (SARS-CoV), and Middle East respiratory syndrome-CoV (MERS-CoV), and SARS-CoV-2 which was discovered in 2019 [4,5,6].

### 1.1. Coronaviruses That Infect the Human Body Respiratory System

Coronaviruses were mostly known to infect animals of mammal species and birds; it is only recently that they have infected humans. The latest research reported seven types of the Coronavirus that can infect humans, four of which are (229E, NL63 HKU1, and OC43). These viruses are reported to cause a respiratory system infection, in which symptoms include coughing, sore throat and coryza. Other reported symptoms that are less likely to occur are pneumonia, bronchiolitis, and bronchitis, whereas three of them MERS-CoV, SARS-CoV and SARS-CoV-2 result in illnesses with fluctuating severity to the human respiratory system. The severity ranges from the common cold to incurable pneumonia [7,8,9,10].

In 2002, severe acute respiratory syndrome (SARS) was reported first in China and soon after affected other countries worldwide. The virus caused infection in human respiratory system organs that were incurable and therefore fatal in most cases [11,12,13]. The virus’s fatality range is 50% higher in seniors than younger adults, with young adults having a ~3–6% fatality percentage [14,15]. A coronavirus known as SARS-CoV was found as the etiological agent of SARS [16].

A decade later a new variation of the virus known as (MERS-CoV) Middle East respiratory syndrome started spreading in the Middle East in 2012; the first fatality reported was at the hospital in Jeddah, Kingdom of Saudi Arabia [17,18,19]. This case was the starting point for a series of infections in surrounding countries and later worldwide [20]. The virus causes lower respiratory system infection and has a 35% fatality percentage. Since the virus outbreak, 2279 cases were reported, 806 of which were fatal. After reports of an outbreak in 27 countries, the WHO listed MERS-CoV as a priority disease requiring urgent research. Notably, this virus’s nature is continuously evolving. Hence, it can efficiently fight human antiviral responses. In retrospect, there are only supportive treatments for the virus [21,22].

In December 2019, seven years after the appearance of MERS-CoV, a new case of virus infection was reported in Wuhan, China. The virus was named by the World Health Organization (WHO) the 2019 novel Coronavirus (2019-nCoV). The name was later changed to SARS-CoV-2 (severe acute respiratory syndrome coronavirus 2), based on its similarities to SARS-CoV. The WHO named the disease caused by the virus infection as COVID-19 [23]. Based on recent data, over 216 M people have reported positive for COVID-19 worldwide by August 2021 [24]. This highly dangerous pandemic has affected all aspects of human day-to-day activities and impeded the most routine activities to a stage that was not possible to predict [25]. In retrospect, international transportations were limited and quarantine and social distancing rules were enforced in most nations [26].

However, the enforced strategies are not a permanent solution as they will have unfavorable impacts on the economy, education, food system including mental wellbeing [27]. In response to the enforced strategies, many have lost their income sources, and others are threatened to lose their job at any minute. It is notable that enforcing strict rules is not controllable and governments are yet to handle this long term [28]. The transmission speed of the virus is soaring and it is a challenge to manage the issues it prevails [29]. This issue is even more crucial when associated with healthcare providers [30]. Having rapid diagnostic technologies is vital to navigating this fatal outbreak [31,32]. Table 1 compares coronaviruses that infect the human body respiratory system; the table compares coronavirus names, year of finding, emergence, type, host, cellular receptor, incubation period, respiratory system infection, symptoms, and mortality rate. The most recently discovered virus in India was the Delta Variant which is a mutation of the SARS-CoV-2 virus. The Delta Variant virus is hosted in humans. The cellular receptor of the virus is ACE2 and the incubation period is 5–6 days. The symptoms accompanied are headaches, sore throat, runny nose, cough, loss of taste and loss of smell.

### 1.2. SARS-CoV and SARS-CoV-2 Genomic Structure and Proteins

The genomic structure and proteins of SARS-CoV and SARS-CoV-2 viruses have similarities more than differences. Both viruses reported being spherical single-stranded RNA viruses that are known to have protein spikes that are protruding from the virion surface. This characteristic in the virus shape gave it its name, Coronavirus, driven from the Latin word corona due to its resemblance to the crown shape [38,39].

SARS-CoV-2 genome consists of 29,903 nucleotides that are significantly similar to SARS such as SARS-CoV that infects bats by 81% nucleotides resemblance. Both viruses have a unique replication approach considering their large RNA genome with none segmented positive-sense RNA genome featuring a shape of 5′cap and 3′ poly (A) tail. The unique structure of the virus allows the replicase polyproteins to understand the genome for translating 2/3 of the genome encodes nonstructural proteins (NSPs). In contrast, the remaining 1/3 encodes for structural and auxiliary proteins. There are four main structural proteins known as (S) spike, (M) membrane, (E) envelope and nucleocapsid (N) proteins. Furthermore, PP1a and PP1b are viral replicas created by the virus are later customized into 16 mature NSPs [40,41]. The pre-mentioned proteins of coronavirus and genomic structure of SARS-CoV and SARS-Cov-2 are illustrated in Figure 2.

The S protein in the virus that weighs about 180 kDa molecular units and is 20 nm long is possibly a target for inhibition of viral entrance and the growth of antibody-based therapeutics to prevent the disease. The protein comprises two subunits which are S1 and S2. The S1 receptor is for binding, which contains the receptor-binding domain (RBD), which identifies and binds to its receptor angiotensin-converting enzyme 2 (ACE2), which is found in the epithelial cells of the lungs. The S2 subunit for membrane fusion, by undertaking specific conformational modifications, induces membrane fusion, which enables viral entry into the host cell. This protein is the chief focus of neutralizing antibodies, which in response prevents infection and further dissemination by stopping binding to ACE2 [42,43].

The E protein is a small membrane protein that manages virion construction and has added effects on the affected host cells. During the host cell infection, the E protein controls the stress response in the cell which, as a result, modifies the virulence of the virus leading to caspase-mediated cell death. The M protein is the amplest in the viral envelope, this protein is essential to the virion particles creation. N protein helps in creating the virial nucleocapsid by forming complexes with the genome RNA within the viral membrane. Both M and N proteins play an important role in RNA replication [44].

## 2. Background of COVID-19 Virus Detection

With the emergence of COVID-19, different detection techniques were developed in the pursuit of containing the rapidly spreading virus by identifying prospective patients using image processing, artificial intelligence, biosensors and others.

Image processing was utilized for the detection of COVID-19; an Image processing technique based on interferometry that uses the phenomenon of interference of waves was developed to detect COVID-19 [45]. Research reported that the computerized tomography (CT) technique used to detect COVID-19 had high accuracy percentage [46]. Another technique based on image processing, Fast COVID-19 Detector (FCOD), was introduced with a promising detection time limit using X-ray images [47]. Infection maps generated by CXR images were used to detect COVID-19 as well as joint localization and severity grading [48]. Additionally, CXR images were employed for the COVID-19 detection via a system called (ACoS) Automatic COVID-19 screening. It uses radiomic texture descriptors by mining CXR images. The system can successfully identify healthy or infected and non-COVID-19 patients [49].

Artificial intelligence has also been widely used in the medical community and, without doubt, was used to detect COVID-19 patients; AI-based techniques were successfully used to diagnose and classify COVID-19 by studying and analyzing X-ray and CT scan images [50]. AI has been helping to prevent the spread of COVID-19 by tracking, diagnosis and social control [51]. CNN deep learning techniques are used to identify potential patients by classifying X-ray images [52]. A decision-making system based on CNN deep learning was developed to assist the radiologist in identifying suspected patients [53].

Deep learning techniques were used to automatically diagnose affected patients which helped radiologists with suspected patients’ diagnoses [54]. A COVNet detection system based on a neural network deep learning model was developed to take out ritualistic characteristics from volumetric chest CT scans for the detection of COVID-19 [55]. A deep learning medical system named (COVIDetction-Net) was developed to automatically identify and detect COVID-19 by using chest radiography images (CRIs) [56]. Popular architectures of deep learning were used to develop a Coronavirus diagnostic system. The architectures are VGG16, DenseNet121, Xception, NASNet, and EfficientNet [57]. A deep convolutional neural network-based architecture was proposed for the COVID-19 detection utilizing chest radiographs [58]. The virus was detected using cough data through artificial intelligence (AI) [59].

An automatic COVID-19 detection approach based on deep learning that applies multilayer-Spatial Convolutional Neural Network CNN was successfully proposed, which can identify affected patients by scanning chest X-ray images and CT scans [60]. Another novel CNN model named CoroDet was developed to detect COVID-19 automatically via CT-scan images and chest X-rays [61]. The deep learning CNN method was also applied to validate and classify chest X-ray images of SARS-CoV-2 suspected patients [62]. Another CNN using profound learning algorithms was proposed to efficiently classify the SARS-CoV-2 virus through CT scans and chest images [63].

Detection techniques merged AI with other approaches for COVID-19 detection; a study presented a joint approach including electrochemical biosensing of SARS-CoV-2 empowered by AI to produce bioinformatics that is needed for an early patient’s diagnosis [64]. A hybrid deep neural network (HDNNs) was developed using computed tomography (CT) and X-ray imaging, to foresee the risk of the onset of disease in patients infected with SARS-CoV-2 [65]. Similarly, a hybrid model joining both deep learning with machine learning was introduced in an effort to arrange possible affected patients’ chest images and classify them as SARS-CoV-2 positive or negative [66]. Additionally, a generative adversarial network (GAN) mixed with in-depth coronavirus detection learning using X-ray chest images was proposed [67]. Another deep learning model that uses a multileveled pipeline was proposed to scan X-ray images to identify the SARS-CoV-2 virus along with other potential chest diseases [68].

Similarly, biosensors were used to detect COVID-19 patients; the MERS-CoV detection approach was developed using an LSPR biosensor with surface-enhanced Raman scattering (SERS) multiplex [69]. Another modern approach was developed called Dual functioning SPR, this approach works by merging the photo-thermal effect in a biosensor LSPR for COVID-19 virus detection [70]. Additionally, a biosensor based on a field-effect transistor (FET) method was developed for the detection and diagnosis of the COVID-19 virus [71]. An overview of the optical biosensors used to identify the COVID-19 virus was published [72]. Researchers successfully developed a COVID-19 detection device based on an optical biosensor that uses SPR with a gold nanoparticle coating [73]. Another Optical biosensor LSPR was demonstrated for the possible detection of coronavirus disease [74]. A rapid and accurate COVID-19 detection using a point-of-care nanophotonic biosensor was developed, it can be used with both human patient samples and animal reservoirs [75]. Promising strategies were proposed for the development of CP-based electrochemical biosensors for COVID-19 detection [76]. An approach of noble metal nanomaterials and associated biosensors was proposed for detecting viruses causing human respiratory system diseases, including COVID-19, using a combination of electrochemical and optical detection techniques [77]. An electrochemical immunoassay was developed for fast and smart COVID-19 detection using samples of saliva [78].

Other strategies for COVID-19 detection were also recorded; a device was developed to detect COVID-19 in a single reaction using a multiples RT-qPCR assay [79]. Another PCR-based optimization using digital droplets was used to identify the COVID-19 virus. This approach exhibited a significantly lower limit of detection compared to RT-PCR [80]. A portable COVID-19 detector tool that is bifunctional electrical with transistors based on graphene field-effect was developed, the tool detects the virus via either acid hybridization or antigen–antibody-protein interaction [81]. An instantaneous and hypersensitive approach to detect anti- SARS-CoV-2 IgG in human serum was developed using lateral flow immunoassay (LFIA) that uses lanthanide-doped polystyrene nanoparticles (LNPs) [82]. On the other hand, a low-frequency Raman (LFR) spectroscopy was used to detect the COVID-19 virus [83].

## 3. Taxonomy of Literature Reviews on COVID-19 Viral Virus

This review studied and analyzed a collection of literature focused on the current viral virus known as COVID-19, which have been divided into two main categories: Transmission COVID-19 Human Exchange and detection techniques of COVID-19 virus. The first category has four subcategories which are Respiratory Transmission, Fecal–Oral Transmission, Ocular Transmission and Vertical Transmission. The second category has two subcategories for detection techniques which are: a. Based on an RNA method that includes next generation sequencing, RT-PCR, LAMP and CRISPR; b. Based on a biosensor that includes electrochemical biosensor, electronic biosensor, piezoelectric biosensor, and optical biosensor as shown below in Figure 3.

The literature was collected using several search engines which were PubMed, WOS, Scopus, Science direct, IEEE Xplore, as well as Google scholar.

## 4. Transmission COVID-19 Human Exchange

With the emergence of the COVID-19 virus, much research has proved that it is transmittable between humans. In order to contain the pandemic, the need to understand the virus’s various modes of transmission is crucial. This section elaborates four human-to-human transmission modes, which are respiratory transmission and fecal–oral transmission. Both are the most popular modes of transmission. Ocular transmission and vertical transmission are also possible modes of transmission; the first one is less common, while the last one is not yet confirmed.

### 4.1. Respiratory Transmission

Collectively the recently discovered coronaviruses which are SARS-CoV, MERS-CoV, and SARS-CoV-2 share the effect of infection in the respiratory tract [84]. Primary SARS-CoV-2 identification was carried out by studying the biofluids of the lower respiratory tract and bronchoalveolar lavage (BAL). The bio fluid was taken from the onset patients present in the Hunan wet market in Wuhan, China on 21 December 2019. All patients had the same symptoms, unknown etiology pneumonia which ranged from the common cold, high temperature, dry coughing to dyspnea. In rare cases, mostly in elderly infected patients, the prementioned symptoms evolved to severe acute respiratory syndrome (SARS), which had similarities to the famous acute respiratory distress syndrome (ARDS) [85]. COVID-19 can transmit from human to human predominantly through respiratory secretions. It can be transmitted either directly through inhalation of the infected human droplets produced from sneezing or coughing; infection transmission can occur through contact by oral, nasal, or ocular mucous membranes, or via contact with surfaces that are infected with human bodily serum. A less possible transmission is via aerosols. The exhaled air produced from the human respiratory system normally contains a high number of droplets that are extremely small. The droplets produced from infected humans will contaminate another human if inhaled or digested [86,87,88]. Previous studies have shown that besides sneezing and coughing, normal breathing and speech can also generate aerosol ranging from 0.75 to 1.1 μm, which is smaller than the aerosol generated by sneezing or coughing (~5 μm) and hence it can transmit the virus up to greater ranges [89]. Another notable transmission method is through airborne dust as it may transport virus-laden through air inhalation causing infection in deeper bronchial and alveolar regions [90]. In the early stages of the current pandemic, the possibility of surface infection with infected droplets being a cause of transmission was a subject of concern to many researchers. Nevertheless, it was proven that it is unlikely to be a method of transmission even though the virus can survive on surfaces for days. Attempts to revive the virus off the contaminated surfaces were in vain. In summary, the most likely transmission of the virus occurs from proportionally large infected droplets produced from infected humans by coughing, sneezing and breathing in close proximity to another person as reported by the infection control guideline [91]. The human exchange of the SARS-CoV-2 virus through respiratory transmission is summarized in Figure 4.

### 4.2. Fecal–Oral Transmission

Although the spread of COVID-19 through droplets, surface contact and aerosolized transmission has been well-publicized, the fecal–oral route is yet another identified method of transmission. According to a meta-analysis by the New England Journal of Medicine, it was found that fecal viral shedding continues throughout the disease, even after nasopharyngeal tests appear negative. Moreover, gastrointestinal symptoms seem to be common for COVID-19 patients, with a prevalence of approximately 18% [92,93]. Research reported that patients who showed symptoms in the digestive system experience significantly longer hospital stays [94]. SARS-CoV-2 and MERS-CoV share 82% genome sequence are reported to trigger respiratory and gastrointestinal (GI) symptoms [95,96]. With that in mind, patients’ stools that are infected with the virus and may be viable under circumstances that can enable fecal–oral transmission; it is possible that COVID-19 could also be transmitted via this route [96]. Fecal–oral transmission occurs through direct or indirect contact with pathogens from contaminated fecal excreta [97]. Holshue et al., (2020) reported case zero of fecal–oral transmission by detecting the patients’ stool RNA of COVID-19 in the US. It was reported that ACE2 is profusely expressed in esophageal, gastric, duodenal and rectal epithelial cells and absorptive enterocytes of ileum and colon allowing the gastrointestinal tract to be prone to COVID-19 infection and therefore suggestive of possible fecal transmission [98,99,100]. Proof of gastrointestinal infection by SARS-CoV-2 was presented by isolating the viral RNA from the epithelial cells and intracellular staining of the nucleocapsid protein [100]. Research in COVID-19 transmission on a family subject in China found two young adults in the same family were diagnosed with diarrhea, proposing that fecal–oral transmission may be a major pathway for virus transmission [36]. COVID-19 virus fecal–oral transmission is illustrated in Figure 5, infected patients pass feces with COVID-19 virus, then the virus is transmitted via unwashed hands, insects and food to other humans.

### 4.3. Ocular Transmission

Recent studies indicate that tears are a potential source for this infection and that the conjunctiva may maintain viral replication for a long period [101]. Another study reported the detection of the SARS-CoV-2 virus in tears of infected patients by RT-PCR [102]. Studies have shown that ACE2 and TMPRSS2 are present in the affected human body and are concentrated in the nasal secretory cells [103]. Several studies have reported high concentrations of ACE2 and TMPRSS2 in the eyes. Immunohistochemical analysis revealed its expression in the conjunctiva, limbus, and cornea [104,105]. With the onset of the SARS-CoV-2 outbreak, conjunctivitis was a rare symptom of the plague. Several patients reported conjunctivitis and other related ocular diseases before the onset of common symptoms of the virus, such as high temperature and dry cough [106].

The human eye could be the entry portal for the virus to enter the body, the eye’s cornea, and conjunctiva form the ocular mucosa surface, which is prone to infection in close proximity with infected people or contaminated hands. Considering the human body anatomy where the eyes mucus is linked to the respiratory system through the nasolacrimal duct, the eyes could be the entry point of COVID-19 [107]. A Chinese expert became infected with COVID-19 and reported the first case of conjunctivitis suggesting a route of conjunctival infection and tear transmission [108].

### 4.4. Vertical Transmission

SARS-CoV-2, resulting in COVID-19 disease, emerged as a third coronavirus epidemic at the end of 2019. Despite the fact that scientific evidence suggests that women are less likely than men to become infected with COVID-19 [109]. Pregnant females were proven to have higher chances of infection with COVID-19 compared to non-pregnant women, which is credited to immunological and anatomical changes. Thus, increasing the risk of developing severe illness [110]. In the case of pregnant females that are infected with the virus, the infection increases morbidity and poses a potential threat to the fetus development due to the spread of COVID-19 receptors known as ACE2 and TMPRSS-2 in fetal organs along with the cells of the maternal–fetal interface [111,112]. In addition, a COVID-19 positive neonate was found in China, the test was confirmed using RT-PCR to test a sample of pharyngeal swabs thirty-six hours after birth; nevertheless, vertical transmission, in this case, was not confirmed [113]. Other risks faced by infected pregnant females are premature birth, intrauterine growth restriction and spontaneous abortion. Yet, there is no strong evidence of COVID-19 virus transmission. Therefore it appears that the threats posed on the fetus are caused by the direct virus effect on the mother. While the current confirmation of vertical transmission during pregnancy is limited, the possible risk of vertical transmission should not be overlooked yet [114,115,116].

## 5. Comparison for Transmission COVID-19 Human Exchange

Transmissions through the human exchange are Respiratory, Fecal–Oral, Ocular and Vertical. Each transmission is compared in Table 2 by probability, possibility, confirmed cases, virus entry organ, transmissibility approach and gender.

Both Respiratory and Fecal–Oral Transmission probability are more likely to happen, they also have a higher possibility of transmission. The entry organs are mouth and nose in Respiratory Transmission via direct and indirect transmissibility approach, while in Fecal–Oral Transmission it is the mouth via indirect transmissibility approach. Both transmissions as well as Ocular transmission can happen for both genders men and women and both have confirmed cases as well as Ocular transmission.

Ocular and Vertical transmission is less likely to happen while Vertical Transmission has the rarest possibility and is yet to be confirmed. The virus entry organ in the ocular transmission is the Eyes via direct and indirect contact whereas in Vertical transmission it is the uterus via direct contact and it can only affect pregnant females.

## 6. Detection Techniques of COVID-19 Virus

The recent pandemic of COVID-19 has drawn attention to the high importance of rapid and accurate diagnostic assays. In parallel to the outbreak, researchers from different areas worldwide have worked together for such assays, concurrently to other many assays that are approved along with the ones that are yet clinically validated. Various techniques of Coronavirus detection have been studied to detect potential patients, in this paper we elaborate on two main methods which are Ribonucleic Acid (RNA)-based techniques and Biosensor techniques.

### 6.1. Based on Ribonucleic Acid (RNA) Method

Molecular diagnostic assays are the leading group of tests used to diagnose COVID-19, mainly using RNA to detect the SARS-CoV-2 virus in patients. The latest detection techniques of COVID-19 RNA, including those still in the development stage, are CRISPR and next-generation sequencing (NGS). Another detection method which is the current golden standard technique is reverse transcription-polymerase chain reaction (RT-PCR) as well as the second most used method which is known as loop-mediated isothermal amplification (LAMP). This section thoroughly explains the previously mentioned techniques and their applications in the efforts of COVID-19 detection.

#### 6.1.1. Next-Generation Sequencing (NGS)

Technologies along with bioinformatics played a significant role in changing the conventional research on viral pathogens and attracting attention to the field of virus diagnosis [117]. Recent research and developments in next-generation sequencing (NGS), alternatively named high-throughput sequencing (HTS) offered to the science field countless applications, and the current outbreak is urging these applications to be wildly used. An advantage of applying NGS for the detection of infectious diseases in the clinical diagnosis is that it is neither culturing nor clinical hypothesis dependent. NGS-based testing discloses all kinds of existing microorganisms in the sample such as fungi, bacteria, parasites and of course viruses, unlike conventional testing methods that require clinicians to do the extra work of addressing patients’ symptoms with possible explanations and requests for testing those particular pathogens [118].

NGS is based on massively parallel sequencing, meaning that billions of short DNA fragments are sequenced simultaneously producing short sequence “reads” rendering dramatically reduced time and cost of sequencing [119]. This technology is reported to be one of the best applications in the current outbreak. Since the emergence of the virus, false-negative results were reported for patients admitted with acute respiratory distress syndrome. It was also reported that this technology was the only one that discovered the etiological pathogen by applying metagenomic RNA sequencing and analyzing the phylogenetic of the complete generated genome allowing us to conclude that the founded new strain of RNA belonged to the Coronaviridae family and was later specified as COVID-19 after nucleotide similarity and genome matching with the existing pathogen’s genome [117]. As a result, NGS technology shows promising results in the efforts of SARS-CoV-2 virus detection.

#### 6.1.2. Reverse Transcription-Polymerase Chain Reaction (RT-PCR)

RT-PCR is the most reliable and gold standard method for the identification of COVID-19 infection [120]. because of its advantages as a precise, and simple quantitative assay [121]. Moreover, real-time RT-PCR showed higher sensitivity compared to RT-PCR assay, which helped greatly in early infection diagnosis [122]. Based on laboratory testing, using this method requires collecting samples for RNA extraction followed by reverse transcription from suspected patients’ upper respiratory tract fluids such as nasal aspirate, nasopharyngeal swab, or pharyngeal swab or the lower respiratory tract (sputum, tracheal aspirate) [123]. After completing the reverse transcription, the cDNA regions are amplified to sufficient levels for pathogen presence detection. This process depends on DNA primer–probe sets complementary to specific regions of the SARS-CoV-2 cDNA, as well as scientists worldwide, who are competing to create these sets since the first SARS-CoV-2 genome was publicly shared. Tests were constructed by finding a number of SARS-CoV-2 particular regions, such as (N/S/E) genes, ORF1ab, and RNA-dependent RNA polymerase (RdRp) [118].

There are two types of RT-PCR testing; the first type uses a single tube containing the required primers to execute the whole process of RT-PCR reaction, the other type is known as two-step, whereby it requires more than one tube to complete the test by doing separate reverse transcription and amplification reactions while offering higher flexibility and accuracy than the other test type. It demands fewer starting materials as well as allowing multiple targets quantification using cDNA stocks. Nevertheless, the first type is more appealing for its quick setup and limited sample handling [124].

Accuracy and specificity are the reason why RT-PCR is widely used despite its expensive price, time-consuming test duration as well as the need for experienced staff [125].

#### 6.1.3. Loop-Mediated Isothermal Amplification (LAMP)

The second in popularity for COVID-19 detection after PCR testing is LAMP testing, PCR testing requires two primers (one in the front and one in the back section), which presents a challenge for LAMP testing that occurs in designing primers and their conflict prevention. Whereas, the LAMP technique needs four or six primers that divide the target sequence into three regions in the forward section and the other three in the back section making a total of six regions. LAMP process is favored because it is executed in an isothermal condition which is an affordable and rapid method. Furthermore, numerous researches reported the application of the LAMP method for coronaviruses detection [126,127,128].

LAMP works on molecular amplification techniques, by rapidly amplifying the genomic material using high efficiency. This technique primarily uses targeted DNA that is synthesized at a stable temperature of 60–65 °C achieved by using explicitly modeled primer sand enzyme (DNA polymerase) that uses strand displacement activity rather than heat denaturation which is used in PCR techniques and within 60 min or less the targeted sequence is amplified to more than 109 copies resulting in the shape of cauliflower including a stem and a loop form of DNA with many inverted repetitions [129].

Researchers combined the techniques of reverse transcription and LAMP (RT-LAMP) and produced a detection method that is characterized by being one-step with the high-throughput method of detection for finding the RNA of SARS-CoV-2 with 100 copies/reaction of an RNA virus. By applying this method the results will be given in 30 min which makes it suitable for POC test and screening application for its simplicity and rapidness compared to RT-qPCR and its needlessness for complicated equipment [130].

#### 6.1.4. Clusters of Regularly Interspaced Short Palindromic Repeats (CRISPR)

Clusters of regularly interspaced short palindromic repeats (CRISPR) possess a frequent series of nucleotides and tiny-scaled spacer sequences. CRISPR-associated proteins known as CAS acts as the nuclease enzymes. Both are used as a bacterial defense system to protect from unknown invaders and are extensively used in RNA modification, therapy using gene alteration, and viral genome detection. Over the last few years, CRISPR has been widely utilized in the field of Vitro diagnostics due to its allele accuracy which is significant in its successful implementation and delivering high accuracy detection and treatment [131].

Several types of research were conducted using CRISPER-based detection systems for SARS-CoV-2 virus detection. For example, a study reported the successful development of a CRISPR-Cas13a-based mobile phone assay that is non-amplified to detect SARS-CoV-2 by testing extracts of RNA taken from nasal swaps. The assay attained ~100 copies/µL sensitivity in less than an hour [132]. Additionally, researches showed detection of SARS-CoV-2 using CRISPR approaches in optimal conditions provides rapid and efficient results. Hence, CRISPR-based approaches are a promising, robust and precise method for SARS-CoV-2 diagnosis [133].

#### 6.1.5. RNA Corona Virus Detection Methods Analysis Based on RT-PCR, LAMP, and CRISPR

This paper studied the detection of Coronavirus using RNA via RT-PCR, LAMP, NGS, and CRISPR Table 3 compares the four techniques by analyzing eleven reported studies. LAMP technique reported fastest detection time of 20–25 min using test sample of Synthetic RNA solution followed by rRT-PCR which takes 30 min using test sample of Plasmids containing the complete N gene. RT-PCR, LAMP, RT-LAMP, and rRT-PCR collectively use Polymerase, (N) Gene, (E) Gene, and ORF1b as target genes, whereas, whereas CRISPR uses (N) and (E) Gene. On the other hand, NGS uses (S) Gene. NGS Detection method has not had much information due to its recent discovery and not many tests were conducted using it.

### 6.2. Based on Biosensor Techniques

Nowadays, researchers are focusing on developing detection tools using biosensors techniques due to their sensitivity, mobility and miniaturization. This section explains four widely used biosensors techniques which are electrochemical biosensors, electronic biosensors, piezoelectric biosensors and optical biosensors.

#### 6.2.1. Electrochemical Biosensors

Electrochemical biosensors are biological concentrations of information that are converted into an analytically relevant signal by using a current or voltage [143]. These biosensing devices can read biochemical information to detect biological materials such as protein and nucleic acid. It also holds qualities such as simple instrumentation, high sensitivity, economic, and the capacity for miniaturization [144,145]. Applications using this type of biosensing device are utilized widely in many areas, whereby it represents a standardized platform for constructing biosensors that include semiconductors and screen-printed electrodes [146]. In brief, these biosensors observe the dielectric changes in properties depicted in dimension, shape, and charge distribution. At the same time, the antibody–antigen complex is formed on the electrode surface, which is categorized into four main groups: potentiometric, amperometric, cyclic voltammetry, and impedimetric transducers [147].

In a recent study focused on SARS-CoV-2 detection, researchers proposed the detection of COVID-19 N-gene using an antifouling electrochemical biosensor. The biosensor is assembled based on electropolymerized polyaniline (PANI) nanowires combined with lately designed peptides. The biosensor detects the N-gene by using biotin-labeled probes that are immobilized onto peptide-coated PANI nanowires, creating an electrochemical interface that is antifouling and susceptible. The recorded detection limit of this biosensor was shallow at (3.5 fm) [148].

#### 6.2.2. Electronic Biosensors

A biosensing device based on field-effect transistors (FETs) consists of a three-electrode structure containing the source, drain, and gate and was developed for the detection of small molecules and the diagnosis of viral diseases. The FET is an electric biosensor that detects changes in surface potential after the target molecule binds to the biorecognition element immobilized on the highly conductive chip surfac [149]. For instance, graphene, zinc oxide, gallium nitride, disulfide, and molybdenum are used in FET-based biosensors where heterogeneous analyte concentrations can be detected rapidly via probes fixed on conducting channels [150]. This type of biosensor is recently used extensively in creating assays for SARS-CoV-2 detection using spike membrane protein] [151]. An example of this method is a graphene FET-based biosensor that can detect COVID-19 related viruses through its spike proteins in only 120 s, also in another way by employing spike protein-specific antibodies, with a 0.2 pM detection limit for the assay [152].

An electronic biosensor device based on FET was developed for the detection of COVID-19 related viruses in clinical samples. The sensor was prepared by covering graphene sheets used in the FET with a unique antibody to the spike protein of SARS-CoV-2. The three determinants of the device’s performance are using a cultured virus, antigen protein, and nasopharyngeal swab specimens from infected patients. The resulting device is a super-sensitive immunological diagnostic method for COVID-19 with minimum sample preparations [71].

#### 6.2.3. Piezoelectric Biosensors

Piezoelectric biosensors are a collection of analytical devices operated based on recording affinity interaction. A piezoelectric platform or piezoelectric crystal is known as a sensor part that works on the basis of oscillations change that results from the mass tied on the piezoelectric crystal surface [153].

Piezoelectric biosensors technology is commonly used to detect hormones, cells, bacteria, viruses, and to study a wide range of interactions on a biomolecular level. This tech offers immediate and unlabelled transduction with high susceptibility, simplicity, and velocity [154,155]. In diagnosing SARS, the piezoelectric biosensor was utilized to measure a type of coronavirus using a sputum sample. To get the antibodies to stick to the piezoelectric crystals, the experiment bound the SARS-CoV horse polyclonal antibody from protein A to the surface. Changes in mass from the crystal, due to viral binding, recorded a shift in frequency [156]. COVID-19 samples are being directly detected by the piezoelectric microcantilever biosensor, which was built without requiring further processing. The biosensor functions as a transducer, and it is coated with an antibody that is relevant to the substance being detected. Because of the mass change caused by the SARS-CoV-2 antigens’ spike proteins, the microcantilever’s surface would experience surface stress and exhibit a quantifiable tip deflection and floating voltage [157]. Compared to other biosensors, piezoelectric biosensors seem to display an enhanced level of performance [158,159]. There is still more study to be carried out before the technology can be applied; while the piezoelectric sensors can detect viral frequency changes, they can also detect them using output voltage directly. On top of that, piezoelectric energy-harvesting devices are anticipated to be used in the IoT, where it is possible to detect viruses by monitoring mechanical vibration [160].

#### 6.2.4. Optical Biosensors

High sensitivity and selectivity are benefits of optical biosensors. They can provide precise detection based on a variety of signals, including absorption, refraction, reflection, dispersion, infrared, polarisation, chemiluminescence, fluorescence, phosphorescence, and so on [161]. In the industry and publications, there are various types of optical biosensors, including fiber-optic biosensors, such as the optrode biosensor and the evanescent wave biosensor, as well as time-resolved fluorescence, the resonant mirror biosensor, interferometric biosensors, and surface plasmon resonance biosensors. They are capable of detecting various biomolecules on both medical and biological specimens, with an impressive window of use [162,163].

Four types of optical biosensors (fluorescence, surface plasmon resonance, localized surface plasmon coupled fluorescence (LSPCF), and fiber optic) will be thoroughly described in this section.

Starting with the up to date known largest group of sensors Fluorescence-based optical biosensors, its popularity is credited to the accessibility of countless fluorescent probes, high quality and fitting optical instruments [164]. This biosensor is characterized by having a variety of intensity, lifetime, energy, transfer and quantum yield that offers opportunities for further exploration [165].

A detection stripe assay based on fluorescent immunochromatographic was built to detect N proteins in a duration of 10 min using samples of nasopharyngeal aspirate and urine with a 68–98% sensitivity rate [166]. Another rapid and quantitative approach of anti-COVID-19 IgG antibody based on the fluorescence biosensor optofluidic POC testing was built. It is an easily handled portable system that is suited for instantaneous results for detecting anti-COVID-19 IgG antibodies in samples. Given perfect conditions, the testing duration can be brought down to 25 min with a detection limit of 12.5 ng/mL that surely meets the diagnostic requirements [167].

Another auspicious platform for pathogens, as well as COVID-19 N gene detection biosensors, is SPR-based biosensors, owing to their characteristic of label-free real-time sensing [151]. An additional label-free SPR-based pioneering aptasensore was developed for COVID-19 N gene detection through thiol-modified niobium carbide MXene quantum dots (Nb2C SH QDs). The resulting tool had a low limit of detection (LOD) of 4.9 pg·mL^−1^. Thus, it exhibited exceptional selectivity in the existence of different respiratory viruses in human bio serum [168].

In addition, the localized surface plasm coupling fluorescence (LSPCF) fiberoptic biosensor was developed to combine a sandwich immunoassay with the SARS coronavirus LSP detection technique. The SARS-CoV N protein can detect very low concentrations (~1 pg/mL) in serum from the LSPCF fiber-optic biosensor [169].

Biosensors that use fiber optics are another type of optical biosensor. A device made from fibers is used in the field of optical science to measure biological species (cells, proteins, DNA, and so on) [161]. To detect the COVID-19 N protein at the point of care, researchers created an easy-to-use plasmonic fiber optic absorbance biosensor (PFAB) that does not need washing. To perform the P-FAB test, the fiber-optic sensor probe is U-bent, and has high EWA (Evanescent Wave Absorbance) sensibility. The COVID-19 N-protein is measured as the light propagates by the U-bent sensor probe linked with the green LED and the photodetector. The P-FAB approach resulted in the lodging of ~2.5 ng/mL or less within 10 min of reading time by using a GOF fused silica/glass optical fiber (GOF) with citrate-capped AuNP labels (size ~40 nm) [170].

## 7. Coronavirus Detection Methods Analysis Based on Biosensor Usage

COVID-19 detection methods based on biosensor usage were thoroughly analyzed by comparing 11 research reported studies from the year 2004 up till now, including the detection of SARS-CoV, MERS-CoV and COVID-19. Table 4 analyzes and compares the development of four biosensor-based techniques with the aim of virus detection. Different materials were used in the biosensors such as PANI (Electrochemical), Gold (Electrochemical), Gold nanoparticles (Optical (P-FAB)), Graphene (Electrical (FET)), Crystal with quartz wafer (Piezoelectric), Nb2C-SH QD (Optical (SPR)) and Polymethyl methacrylate (Optical (LSPCF)). Detection of COVID-19 using electrochemical biosensor reported the lowest temperature of 4 °C and fastest detection time of 10–30 s using test samples of spiked saliva.

## 8. COVID-19 Detection Techniques Advantages and Limitations

The detection techniques that were extensively explained previously are like any other techniques have their advantage and limitations. RNA-based detection techniques (RT-PCR, LAMP, NGS, and CRISPR) advantages include specificity and sensitivity. In comparison, limitations include the need for extracting RNA from clinical samples and the inability to detect already recovered patients. On the other hand, biosensors advantages include affordability for both electrochemical and electronic sensors, whereas for piezoelectric and optical advantages include rapidness, specificity and sensitivity. The common limitation for electrochemical and electronic is that both are time-consuming. Below, Table 5 illustrates both aspects of each detection technique.

## 9. Conclusions

The COVID-19 pandemic is ongoing globally and is yet to be contained; simultaneously, research in enhancing methods of preventing the spread is persistent. This review article compiled and studied research concerning the recently discovered COVID-19 virus that is rapidly morphing and highly transmittable between humans. This article addressed the pre-mentioned issue by comparing and analyzing SARS-CoV-2 transmission methods between humans and detection methods utilized to detect COVID-19. Human-to-human exchange transmission through the respiratory system proved to be the most efficient method of transmission, followed by fecal–oral transmission. Rare cases were recorded for the ocular transmission method, whereas vertical transmission has not yet been confirmed. In synchronization with the virus’s rapid spread, advancements in the detection techniques with effective diagnoses were urgently needed by clinicians. This paper gives a comprehensive review of COVID-19 virus detection techniques based on RNA and biosensors. RNA-based methods included the golden standard testing set by WHO known as RT-PCR and RT-LAMP. The latter is deemed a possible proper substitute due to it being more economical and rapid compared to RT-PCR. Last but not least, the paper reviewed the latest advancements in detection techniques using biosensors for SARS-CoV-2 detection, by using target genes in clinical samples which are the Spike (S) protein, Nucleocapsid (N) protein as well as Antigen sputum, and IgM antibody. Biosensors are proven to be rapid, sensitive, precise, mobile, economical, and have the ability for miniaturization. Therefore, COVID-19 detection using biosensors is innovative and promising.

## Figures and Tables

**Figure 1 sensors-21-08362-f001:**
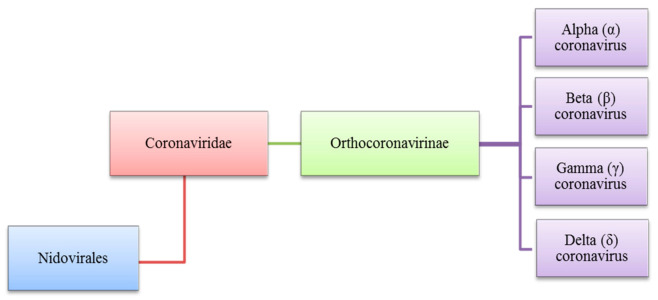
Diverse types of coronaviruses within Nidovirales, Coronaviridae family, Orthocoronavirinae subfamily and the respective genera, (α), (β), (γ) and (δ).

**Figure 2 sensors-21-08362-f002:**
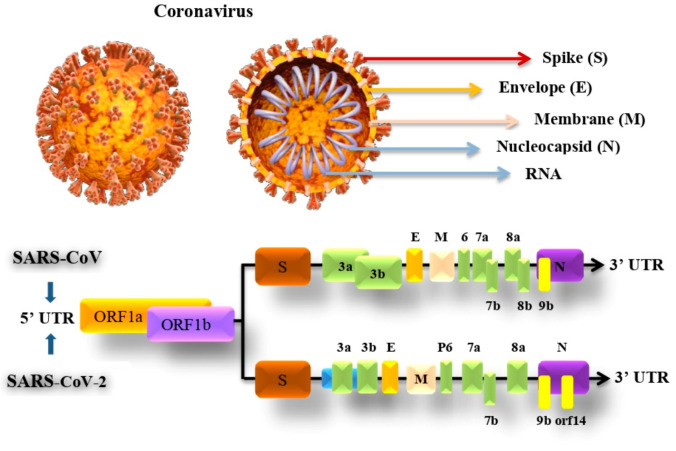
SARS-CoV and SARS-CoV-2 genomic structure and proteins.

**Figure 3 sensors-21-08362-f003:**
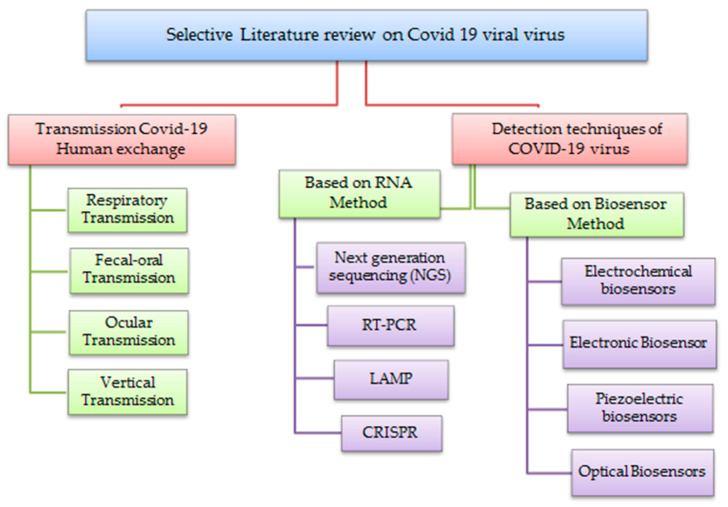
Taxonomy of literature reviews on COVID-19 viral virus.

**Figure 4 sensors-21-08362-f004:**
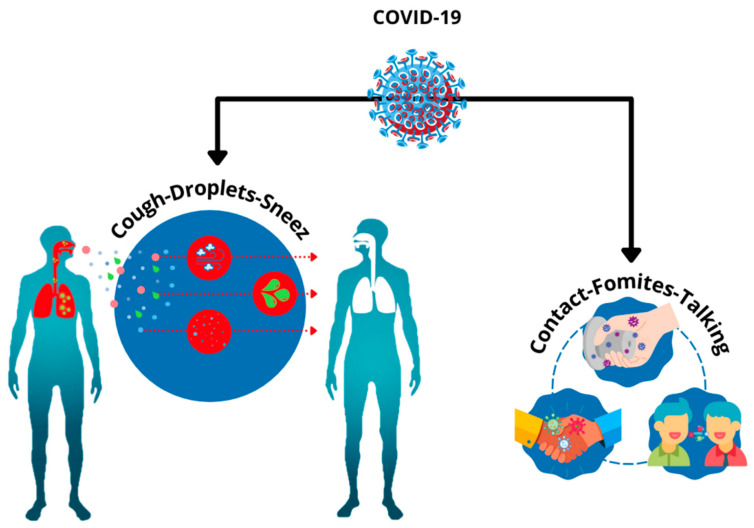
COVID-19 respiratory transmission Human-to-Human.

**Figure 5 sensors-21-08362-f005:**
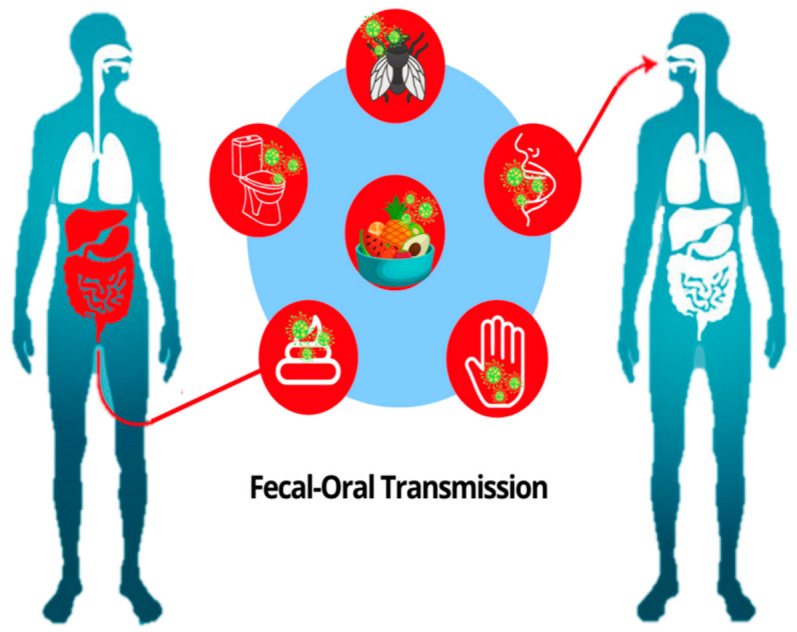
Fecal–oral COVID-19 transmission.

**Table 1 sensors-21-08362-t001:** Comparative Coronaviruses that infect the human body respiratory system.

Coronaviruses	Year of Finding	Emergence	Type	HOST	Cellular Receptor	Incubation Period	Respiratory System Infection	Symptoms	Mortality Rate	Reference
HCoV-HKU1	2005	Hong Kong	Beta	Human	9-O-Acetylated sialic acid	2–4 days	√	Common cold, Bronchitis, and pneumonia.	N.A.	[33]
HCoV-Nl63	2004	Holland	Alpha	Human	ACE2	2–4 days	√	Common cold, sore throat, bronchiolitis/croup in children, high temperature, malaise, coughing and rhinitis	N.A.	[34]
HCoV-229E	1966	N.A.	Alpha	Human	Human aminopeptidase N (CD13)	2–5 days	√	Common cold, Headache, Fever, Running nose, Pneumonia (in neonates), malaise, Bronchiolitis,	N.A.	[35]
HCoV-OC43	1967	N.A.	Beta	Human	9-O-Acetylated sialic acid	2–5 days	√	Running nose, Common cold, Fever, Headache, Malaise, Bronchiolitis, Pneumonia (in neonates)	N.A.	[35]
SARS-CoV	2002	Guangdong, southern China	Beta	Civets, Human	ACE2	2–11 days	√	Headache, Diarrhea, Fever, Shivering, Dyspnea, Cough, Pneumonia, Myalgia	10%	[33]
MERS-CoV	2012	Jeddah, Saudi Arabia	Beta	Camel, Human	DPP4	2–14 days	√	Fever, sore throat, dyspnea, dry cough, Chills, Pneumonia, Myalgia, Diarrhea, Hemoptysis, Headache, Rhinorrhoea	35%	[19]
SARS-CoV-2	2019	Hubei province, Wuhan city, China	Beta	Bat, Human	ACE2	3–6 days	√	High body temperature, Short of breath, Headache, Sore throat, myalgia, Dry coughing, Anosmia, rarely pneumonia, Diarrhea, Generalized weakness, Nasal congestion, Rhinorrhea, Sneezing	2.8%	[36]
Delta Variant *	2020	India	Delta	Human	ACE2	5–6 days	√	Headaches, Sore Throat, Runny Nose, Replacing Cough and Loss of Taste, Loss of Smell	N.A.	[37]

NA = Not Available, * Delta Variant is a mutation of SARS-CoV-2.

**Table 2 sensors-21-08362-t002:** Human exchange transmission comparison.

Comparison	Transmission Human Exchange			
	Respiratory Transmission	Fecal–Oral Transmission	Ocular Transmission	Vertical Transmission
Probability	More likely	More likely	Less likely	Less likely
Possibility	High	High	Rare	Rarer
Confirmed Cases	Confirmed	Confirmed	Confirmed	Unconfirmed
Virus entry organ	Mouth, Nose	Mouth	Eye	Uterus
Transmissibility approach	Direct, Indirect	Indirect	Direct, Indirect	Direct
Genus	Male and Female	Male and Female	Male and Female	Female

**Table 3 sensors-21-08362-t003:** RNA Coronavirus detection methods analysis based on RT-PCR, LAMP, NGS and CRISPR.

Reference	Coronavirus	Analyte	Target Genes	Detection Methods	Limit of Detection	Concentration Range	Detection Time	Tested Sample
[134]	SARS-CoV	RNA	Polymerase	RT-PCR	10 copies/reaction	N.A.	(5) h	Nasal aspirate
[135]	SARS-CoV	RNA	NA	RT-PCR	2 nM	N.A.	(~2) h	Throat swab samples
[136]	MERS-CoV	RNA	(N) gene	rRT-PCR	10 copies/reaction or 0.0013 TCID50/ml	10–10^8^ copies/-reaction	(~2) h	Serum, nasopharyngeal/- oropharyngeal swab, and sputum samples
[137]	COVID-19	RNA	(E)-gene	rRT-PCR	275.7 copies/reaction	N.A.	(~1) h	Swab samples
[138]	COVID-19	RNA	(N) gene	rRT-PCR	10 copies/reaction	N.A.	(~30) min	Plasmids containing the complete N gene
[139]	MERS-CoV	RNA	(N) gene	RT-LAMP	10 copies/μL	5 × 10^1^–5 × 10^8^ copies/-reaction	(35) min	Throat swab specimens
[140]	SARS-CoV	RNA	(ORF1b) and (N) gene	LAMP	10^4^ copies/reaction	N.A.	(20–25) min	Synthetic RNA solutions
[141]	COVID-19	RNA	(ORF1b) and (N) gene	RT-LAMP	20 copies/reaction	N.A.	(20–30) min	Nasopharyngeal swab and bronchoalveolar lavage fluid samples
[126]	COVID-19	RNA	(ORF1ab), (N) and (E) gene	RT-LAMP	5 copies/reaction	N.A.	(30) min	Nasopharyngeal swab specimens
[142]	COVID-19	RNA	(S) gene	NGS	125 GCE/mL	N.A.	N.A.	Nasopharyngeal swab
[132]	COVID-19	RNA	(N) and (E) gene	CRISPR/Cas13a	~100 copies/µL	3.2 × 10^5^ –1.65 × 10^3^ copies/µL	(30) min	Nasal swab

NA = NOT available, N = nucleocapsid, E = envelope, ORF1b = open reading frame 1b, S = s pike protein.

**Table 4 sensors-21-08362-t004:** Coronavirus detection methods analysis based on biosensor application.

Reference	Publication Year	Coronavirus	Biosensor Detection Technique	Material	Target	Detection Time	Linear Range	Tested Sample	Limit of Detection	Temperatures
[148]	2 April 2021	COVID-19	Electrochemical	(PANI)	N gene	1 h	10^−14^ to 10^−9^ M	NR	3.5 fM	37 °C
[171]	11 May 2020	COVID-19	Electrochemical	Gold	S protein	10–30 s	1 fM to 1 μM	Spiked saliva samples	90 fM	4 °C
[172]	27 February 2019	MERS-CoV	Electrochemical	Gold	S protein	20 min	1 pg·mL^−1^ to 10 μg·mL^−1^	Spiked nasal samples	0.4 and 1.0 pg·mL^−1^	RT
[71]	15 April 2020	COVID-19	Electrical (FET)	Graphene	S protein	4 h	NR	nasopharyngeal swab	1.6 × 101 pfu/mL	NR
[152]	2020	COVID-19	Electrical (FET)	Graphene	S protein	2 min	NR	Spiked spike protein solutions	0.2 pM	NR
[156]	1 July 2004	SARS-CoV	Piezoelectric	Crystal with quartz wafer	Antigen sputum	1 h	1–4 µg/µL	NR	0.60 mg/mL	RT
[166]	13 March 2020	COVID-19	Optical (fluorescence)	Not Specified	N protein	10 min	NR	Nasopharyngeal aspirate swabs and urine	Not Specified	NR
[167]	14 August 2021	COVID-19	Optical (fluorescence)	Not Specified	IgG	25 min	NR	Human serum	12.5 ng/mL	NR
[168]	11August 2021	SARS-CoV-2	Optical (SPR)	Nb2C-SH QD	N gene	NA	0.05 to 100 ng·mL^−1^	Human serum	4.9 pg·mL^−1^	NR
[169]	17 July 2009	SARS-CoV	Optical (LSPCF)	polymethyl methacrylate	N protein	2 h	0.1 pg/mL to 1 ng/mL	Human serum	∼1 pg/mL	37 °C
[170]	1 September 2021	COVID-19	Optical (P-FAB)	Gold nanoparticles	N protein	10 min	0.1 ng/mL and 100 ng/mL	PBS Buffer	~2.5 ng/mL	NR

Electropolymerized polyaniline (PANI) nanowires, (G/SPCE) = A graphene-modified screen-printed carbon electrode, RT = Room Temperature, (P-FAB) = plasmonic fiber optic absorbance biosensor, NR = Not Reported, N = nucleocapsid, S = spike protein.

**Table 5 sensors-21-08362-t005:** COVID-19 detection techniques advantages and limitations comparison.

Categories of Coronavirus Detection	Techniques	Advantages	Limitation
Indirect detection RNA	RT-PCR	❖ Reliable/detects current viral infection ❖ Rapid results ❖ Higher sensitivity ❖ Needs a slight amount of DNA ❖ It can be performed in a single step ❖ Well established methodology in viral diagnostics	❖ Inability to detect already recovered patients. ❖ Depends on Spiked-in material. ❖ Requires sophisticated instruments. ❖ Low sensitivity. ❖ The need to extract RNA. ❖ Higher costs due to expensive consumables usage. ❖ Complex detection.
Indirect detection RNA	LAMP	❖ Extremely repeatable ❖ Precise ❖ One working temperature (60–65°) ❖ Rapid reaction ❖ Accurate ❖ Fast amplification ❖ Operation Simplicity ❖ Detection Simplicity	❖ Low versatility ❖ Possibility of primer–primer interactions ❖ Low detection sensitivity. ❖ Long detection time. ❖ The need to extract RNA from clinical samples. ❖ Cannot detect recovered patients. ❖ Too sensitive, highly susceptible to false ❖ positives due to carry-over or cross-contamination
Indirect detection RNA	NGS	❖ Highly sensitive ❖ Specific ❖ Can identify the novel strain	❖ Slow detection. ❖ The need to extract RNA from clinical samples. ❖ High expertise required ❖ Equipment dependency ❖ Expensive ❖ Requires highly sophisticated Lab ❖ Unable to detect recovered patients.
Indirect detection RNA	CRISPR	❖ Affordable ❖ High detection speed ❖ High sensitivity	❖ Several CRISPR-based kits are still in development phase ❖ Clinical validation is required ❖ Not yet widespread and in clinical trials
Indirect detection: Spike (S) and Nucleocapsid (N) proteins	Electrochemical sensors	❖ Structure simplicity ❖ High-level sensitivity ❖ Economical ❖ Instant response ❖ Label-Free	❖ High need for kits. ❖ Tiresome sample collection. ❖ Time-consuming process ❖ Requires trained personnel ❖ Requires adequate laboratory infrastructure
Indirect detection: Spike (S) protein	Electronic sensors(FET)	❖ Low cost ❖ High speed ❖ Small size ❖ Excellent compatibility with integrated circuits (ICs).	❖ The signal transduction process found is not always apparent. ❖ Heterogeneous interface structures. ❖ Long time result.
Indirect detection: Antigen sputum	Piezoelectric sensor	❖ Rapid ❖ Highly sensitive ❖ Specific ❖ Label-free	❖ Depends on sample preparation ❖ Complex pretreatment steps
Indirect detection: IgM antibody and Nucleocapsid (N) Protein	Optical (fluorescence, SPR, LSPCF, and P-FAB)	❖ Rapid ❖ Sensitive ❖ Specific	❖ Requirement for point of care remains difficult. ❖ High cost.

## Data Availability

This manuscript has not been previously released and is not now under consideration by any journal for publication.

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
