# Peer review of "Human Body Performance with COVID-19 Affectation According to Virus Specification Based on Biosensor Techniques"

_sensors, 2021, doi:10.3390/s21248362_

Round 1

Reviewer 1 Report

The manuscript by Mohammed Jawad Ahmed Alathari et al. would be of interest to the readers who want to know the basis of COVID-19 and its diagnosis methods. Common modes of virus transmission have also been reviewed. However, I do not think it is acceptable for Sensors at its current stage.

Major issues:

  1. The detection of COVID-19 and the virus have been extensively reviewed during the last two years. The authors have to emphasize the key contribution of this manuscript in the abstract, in comparison with previously published review articles.

  1. Too much content is out of focus. Some basic and irrelevant introduction is not necessary to a topic under sensors or biosensors. The recent developments about COVID-19 detection are not well summarized. The authors have to delete unnecessary contents and extend those concerning with detection or diagnosis.

  1. Although the authors have introduced some detection techniques that have been applied for COVID-19 diagnosis, but most of the contents are the basic knowledge and general information of those techniques. There is lack of specific discussion toward COVID-19 detection. E.g, it’s better for the authors to review how to design the primers for COVID-19 virus specific detection using PCR-based assays? How are the sensitivity and specificity of these methods? What are the advantages and potential limitation of each detection method?

Other issues:

  1. Please uniform the formats of all tables, e.g. the reference citation was found in the first row of table 3 but in the second row in table 4.
  2. Please correct the format of references, the Journal’s name is missing.
  3. Please be noticed that Wuhan is the first place that had reported the disease caused by COVID-19 virus, but not the origin of the virus. This is supported by WHO. Please recheck the entire manuscript to make the description correct.

Author Response

Dear Reviewer, 

We want to thank you for your time and efforts. Your comments helped create a better version of the manuscript. 

We have attached the responses for your kind reference.

Reviewer 2 Report

This paper shows an overview of the Covid-19 pandemics starting with the virus spreading, virus structure and etiology, transmission to end up with different detection techniques implemented with more or less sucess to aid in the early detection of the virus infection. Even though the paper provides a thorough review on these topics, there are some flaws the authors should thoroughly amend and discuss accordingly. These are listed below.

  1. English style should be corrected. Some sections contain many grammatical errors that could easily be worked out.
  2. The virus is named after Covid-19 or SARS-CoV-2 thoughout the paper. The former is the name of the disease while it is the latter the official name for the virus. This should be corrected and I suggest paying attention to these terms that could be misleading.
  3. Line 109 - it refers to the Delta variant as if it were a different virus and it is just a mutation of the same virus. Please, make sure that all these terms and definitions are cited correctly.
  4. Line 134 - ACE2 enzyme is not found in the human lungs but in the epithelial cells of the lungs.
  5. Lines 182-183 is an example of a typographical error of a sentence that does not make sense.
  6. Lines 217-218 - What is the difference between an electrochemical and an electronic sensor? Later in section 6, the authors name FETs as electronic sensors when they are potentiometric sensors being a type of electrochemical sensors. This should be corrected thoroughly.
  7. The same applies to piezoelectric sensors. Piezoelectricity is the property used to measure changes in mass upon interaction with an analyte and so the sensors are mass sensors. The classification give for biosensors is all wrong!.
  8. Section 6.1. There are also biosensors based on the detection of specific sequences of the viral RNA.
  9. Sections 6.1. And what about CRISPR and Sherlock techniques?
  10. NGS is not a high thoughput screening technique and as such has not been used for massive detection. Including this technique here is a bit risky.
  11. How do the authors state that RT-PCR is not suitable for virus detection when it has been the gold standard technique and massively used across the world?
  12. Section 6.2.1. This section should be thoroughly rephrased. The definition of an electrochemical biosensor is not correct and all is mixed up.
  13. As I mentioned in point 6, FETs are electrochemical (potentiometric) biosensors and so, sections 6.2.1 and 6.2.2. should be merged.
  14. Line 475 and 476. What is LSPCF? A fiber optic biosensor could be a fluorescent or an absorbance biosensor, too? This part should be corrected paying attention to the classification of optical biosensors.
  15. Please, re-write the abstract and conclusion sections after amending the paper following the previous comments.

Author Response

(The authors gave the same response as above.)

Round 2

Reviewer 1 Report

There is no more comment.

Author Response

Thank you for your time and efforts, we hope that the edited version meets your high standards.